# A Study on State of Charge and State of Health Estimation in Consideration of Lithium-Ion Battery Aging

**Woongchul Choi**

Department of Automotive Engineering, Kookmin University, Seoul 02707, Korea; danchoi@kookmin.ac.kr

**Abstract:** Due to rapid development of industries around the world, more and more consumption of fossil fuels was unavoidable, resulting in serious environmental problems. The many pollutant emissions—a major contributor to global warming and weather pattern change—have been at the center of concern. In order to solve this issue, research and development of electric vehicles and energy storage systems made great progress and successfully introduced products in the market. Nevertheless, accurate measurement of the state of charge (SOC) and state of health (SOH) of the Li-ion battery, the most popular electric energy storage device, has not yet been fully understood due to the nature of battery aging. In this study, ideas to estimate the capacity and ultimately SOC and SOH of Li-ion batteries are discussed. With these ideas, we expect not only to accommodate the issues with battery aging but also to implement an algorithm for an on-board battery management system. The key idea is to chase and monitor internal resistance continuously in a fast and reliable manner in real time. With further investigation of the key idea, we also fully expect to come up with a reliable SOC and SOH measurement scheme in the near future.

**Keywords:** Li-ion battery; battery capacity; state of charge (SOC); state of health (SOH); aging; battery management system (BMS)

## 1. Introduction

The rapid development of technology and industrialization has triggered heavy use of fossil fuels and has resulted in serious environmental problems, including pollutant emissions and global warming [1–3]. Thus, the research and development of eco-friendly energy utilization technology has been rapidly progressing [4,5]. Of the many industrial areas, including power generation, manufacturing, housing, and transportation, the transportation sector has been identified as one of the major responsible parties for pollutant emissions and global warming [6]. Therefore, research and development in electric vehicles (EVs) has been accelerating [7]. Among the many EV-related research projects, energy storage devices (i.e., batteries) have attracted major attention since batteries play important roles not only in EVs but also in mobile devices, personal transportation, energy storage systems (ESS) for renewable energy generation, and more. Naturally, research on batteries has been accelerating as well [7–9]. In addition, a battery management system (BMS), which is used for the efficient operation and management of batteries, is also considered as an important system in various applications requiring high energy storage capacity, including electric vehicles and energy storage systems [10–12].

With the main focus on EV applications, a forecast of future EV sales is shown in Figure 1, based on data from EV Outlook and announced by BloombergNEF [13]. With this increase in EV sales, it is natural to expect that the number of EVs retired from service will increase in a similar manner with a time delay of 10 or more years. With the recent development of the Li-ion battery, especially for the

case of EVs, battery life is actually quite long as long as a battery is used properly. So, short battery life is the main issue. The main focus of the current research is to come up with a new methodology to diagnose battery capacity, typically denoted as $Q_{max}$, when a battery is returned for second use after an EV retires. For example, conventional diagnostic methods take quite a long time—up to 10 h or even more. However, the newly proposed methodology would complete the test within 30 min or less. With this fast diagnostic capability, the battery reuse process will make sense in terms of being a profitable business activity, and as a result, a battery resource can be reused as part of a self-sustaining business model. Without fast diagnostic techniques, the second use of batteries can only be discussed but cannot be realized. Furthermore, only with this updated battery capacity can state of charge (SOC) and state of health (SOH) be estimated accurately as a battery ages more and more. Therefore, a newly proposed method for the fast measurement of battery capacity is the main contribution of the current research.

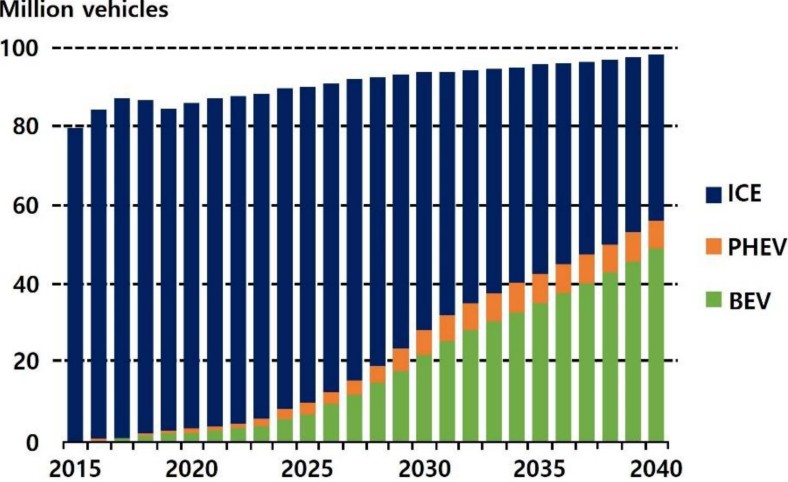

**Figure 1.** Global long-term passenger vehicle sales by drivetrain [13].

## 2. Methodology Development

A newly proposed idea for the fast estimation of battery capacity was developed by adopting the internal resistance tracking method. After the establishment of the initial idea, an equivalent circuit model of a specific Li-ion battery was established based on a Shepherd model that would mimic open circuit voltage (OCV) characteristic behaviors very well [14]. The main reason for constructing the battery model was to produce numerous simulation data for the successful implementation of the new methodology. With the established model, multiple simulations were carried out to produce quantitative data that clearly showed the increase in internal resistance and the decrease in battery capacity as the battery aged. With the help of simulation data, the newly proposed approach was refined, and after the successful implementation of the new methodology, real battery experiments were conducted to prove the concept of fast measurement of battery capacity. In the following sections, conventional methods and the newly proposed method are explained.

### 2.1. Conventional Methods for SOC Measurement

Typical SOC measurement techniques include a voltage-based estimation method, a coulomb counting estimation method, and an internal resistance tracking method [15–23]. Since the first and the second methods are quite commonly used in the field even with the lack of consideration of a battery aging effect, they are not detailed in this paper. However, the third technique is explained in detail as it is the basis for the newly proposed method.

The internal resistance tracking method consists of using battery capacity estimation, the voltage-based method, and the coulomb counting method together to track internal resistance and

eventually to calculate the SOC and SOH of the battery, considering the aging effect [23]. As shown in Figure 2, a discharging process is illustrated to explain the basic idea. After the initial discharging process of any shape (from 0 s to 2000 s), the battery goes through the first relaxation period. After reasonable relaxation, the OCV at that time is measured and denoted as OCV1. Then, the SOC of the battery is obtained using the voltage-based method and denoted as SOC1. After that, a second discharging process is carried out. During the second discharge, the amount of discharged energy is measured with the coulomb counting method and denoted as $\Delta Q$. Then, the battery goes through a second relaxation period for the measurement of OCV and is denoted as OCV2, resulting in the estimation of SOC using the voltage-based method and is marked as SOC2, eventually.

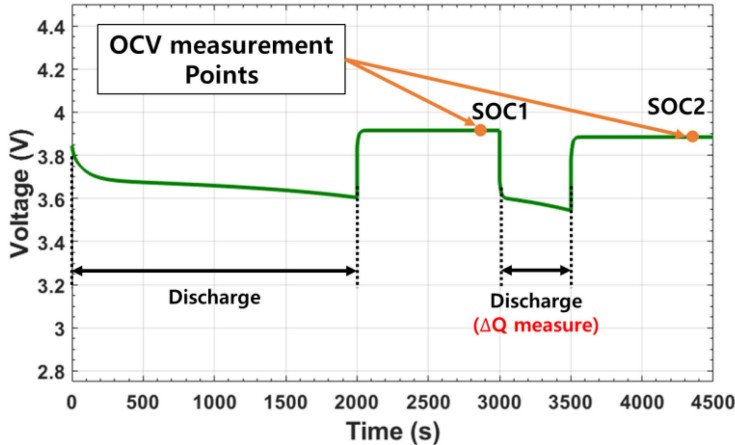

**Figure 2.** Illustration for the internal resistance tracking method.

With the successful measurement of SOC1 and SOC2, along with $\Delta Q$, the battery capacity, denoted as $Q_{max}$, can be estimated by plugging values into Equation (1), which is derived from the definition of state of charge [23].

$$Q_{max} = \frac{\Delta Q}{(SOC1 - SOC2)} \tag{1}$$

With the result from Equation (1), $Q_{max}$ can be compared to the initial quantity at the beginning of life (BOL). As the battery experiences aging, $Q_{max}$ decreases as a result [20,21]. An effective metric of the SOH can be established quickly, such as $Q_{max}$ (now)/$Q_{max}$ (BOL). The only issue with this concept is a rather long relaxation period, which may take about an hour or more for stable OCV measurement.

Internal resistance, which is another important battery characteristic, can be estimated as the battery ages based on the voltage drop between the OCV and the instantaneously measured voltage, denoted as $V_{inst}$. Instantaneous voltage during the discharge process can be modeled by Equation (2), as shown below, where T is for temperature, $I_{inst}$ is for instantaneously measured current, and R is for resistance.

$$V_{inst} = OCV(T, SOC) - I_{inst}R(T, SOC, Aging) \tag{2}$$

Therefore, voltage drop, $\Delta V$, which is equal to $OCV(T,SOC) - V_{inst}$, can be measured and saved in real time. At the same time, a series of estimated internal resistance values can be calculated simply by Equation (3).

$$R(T, SOC, Aging)_{inst} = \frac{\Delta V_{inst}}{I_{inst}} \tag{3}$$

During this process, SOC values are obtained based on the voltage-based method, which is valid with $Q_{max}$ (BOL). However, in the internal resistance tracking method, $Q_{max}$ (now) is used to adjust the SOC value, accounting for aging effect, and therefore the SOC value can be estimated more accurately. Along with the tracked resistance, the updated $Q_{max}$ value plays an important role in determining the remaining usable energy as it reflects the true status of the battery, namely, the aged resistance and the

aged battery capacity at that moment. Additionally, with the continuously updated battery capacity $Q_{max}$ (now), SOH can be represented as $Q_{max}$ (now)/$Q_{max}$ (BOL) since the capacity decreases as the battery ages. The only difficulty with this method is the long battery relaxation time needed to obtain $Q_{max}$ (now).

Regarding conventional SOH estimation techniques, the most popular choice among researchers is electrochemical impedance spectroscopy (EIS), which measures the impedance of the battery by injecting a small amount of alternate current (AC) into the battery over a wide range of frequencies, typically from 0.1 Hz to 1000 Hz [24,25]. However, none of these methods can be implemented in on-board devices for real-time measurements because EIS requires an off-line setup for the expensive EIS measurement system, and the existing internal resistance tracking method requires a long relaxation time for satisfactory voltage stabilization.

*2.2. Newly Proposed Internal Resistance Tracking Method*

In Equation (1), many researchers paid close attention to obtaining both SOC1 and SOC2 separately. For this matter, it usually takes a long relaxation time to obtain stable SOC1 and SOC2 independently. However, in the current research, a new approach is introduced to overcome this shortcoming. In principle, the difference of SOC1 and SOC2 was directly obtained rather quickly and without long relaxation time periods. As shown earlier in Figure 2, with accurate OCV readings, $Q_{max}$ can be estimated quite accurately from Equation (1). However, as illustrated in Figure 3, so-called accurate OCV readings can be obtained only after a long relaxation time period, as noted by yellow arrow marks, and then the corresponding SOC1 and SOC2 can be estimated from the SOC–OCV curve. After that, $Q_{max}$ can be calculated from Equation (1). In the illustration provided in Figure 3, OCV1 readings after 3500 s and OCV2 readings after 3500 s are shown, and the corresponding SOC1 and SOC2 after 3500 s are noted. As mentioned earlier, with this conventional internal resistance tracking method, it is almost impossible to assess $Q_{max}$ for an on-board BMS system due to the long relaxation time.

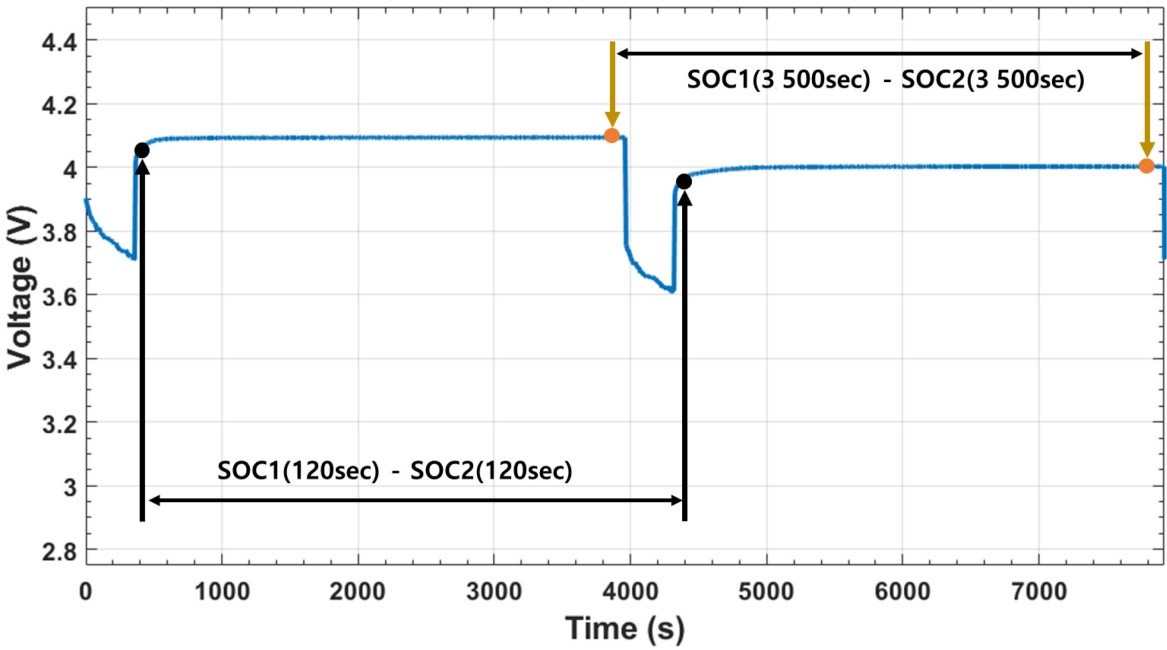

**Figure 3.** Illustration for the newly proposed internal resistance tracking method.

In the newly proposed approach, the following concept is introduced. Since the important value for successful estimation of $Q_{max}$ is the difference between SOC1 and SOC2, namely SOC1–SOC2, not the absolute values of SOC1 and SOC2, new measurement points are taken. The estimation of SOC1–SOC2

from the respective OCV values are obtained after a reasonably short time instead of waiting for a long relaxation time. In Figure 3, it is noted in black arrows and, more specifically in this case, the SOC1–SOC2 value is estimated after only 120 s. As a proof of concept for the proposed approach, simulated battery data from an equivalent circuit model were heavily used during the establishment of the new methodology. After the satisfactory results from the initial algorithm development, extensive experimental data were collected to further refine the new methodology.

## 3. Battery Simulation Based on the Shepherd Model

An equivalent circuit model for the simulation of lithium-ion battery behavior was established based on the Shepherd model [14]. One of the popular cylindrical types of lithium-ion battery, a Samsung 18650-26F, Samsung SDI, Seoul, Korea, was selected for the simulation since the actual battery was readily available for experimental analysis. The specifications of the target battery are provided in Table 1 and the capacity and discharge voltage of the Samsung 18650-26F were used to complete the Shepherd model as shown in Figure 4 [26].

**Table 1.** Lithium-ion battery specifications.

| Nominal Capacity | | 2600 mAh |
|---|---|---|
| Minimum capacity | | 2550 mAh |
| Nominal voltage | | 3.7 V |
| Discharge cut-off voltage | | 2.75 V |
| Charging voltage | | 4.2 V |
| Calculated internal resistance | | 0.09 Ω |
| Max charge and discharge current | Charge | 2600 mA |
| | Discharge | 5200 mA |
| Operating temperature | Charge | 0~45 °C |
| | Discharge | −20~60 °C |

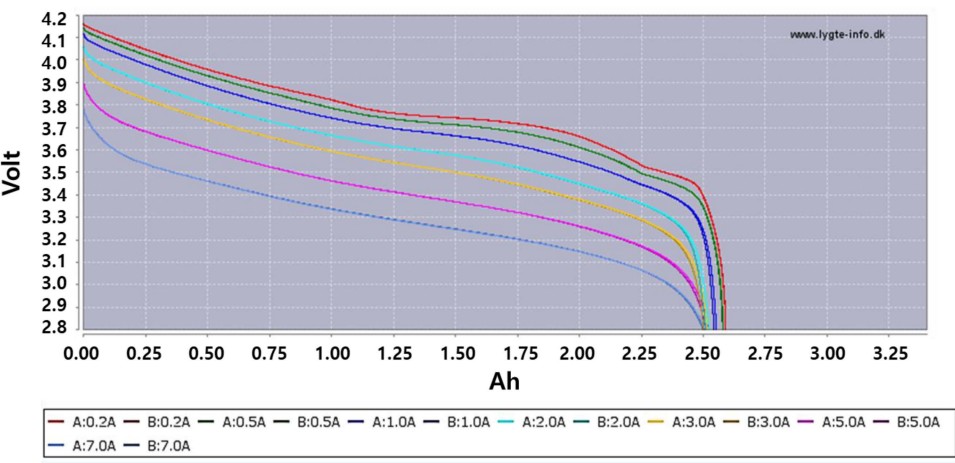

**Figure 4.** Capacity and discharge voltage curve of Samsung ICR18650-26F.

During the implementation of the internal resistance tracking method, simulation data from the Shepherd model were heavily used instead of using the experimental data. The battery model was developed in the Matlab Simulink environment, as briefly shown in Figure 5.

The necessary parameters in the Shepherd battery model were obtained through the capacity and discharge voltage curve [26]. In case of internal resistance, it was assumed in the model that it increased rather linearly in each cycle of 0, 25, 50, 75, and 100, and, consequently, the internal resistance in the model increased about 1.2 times when reaching 100 cycles.

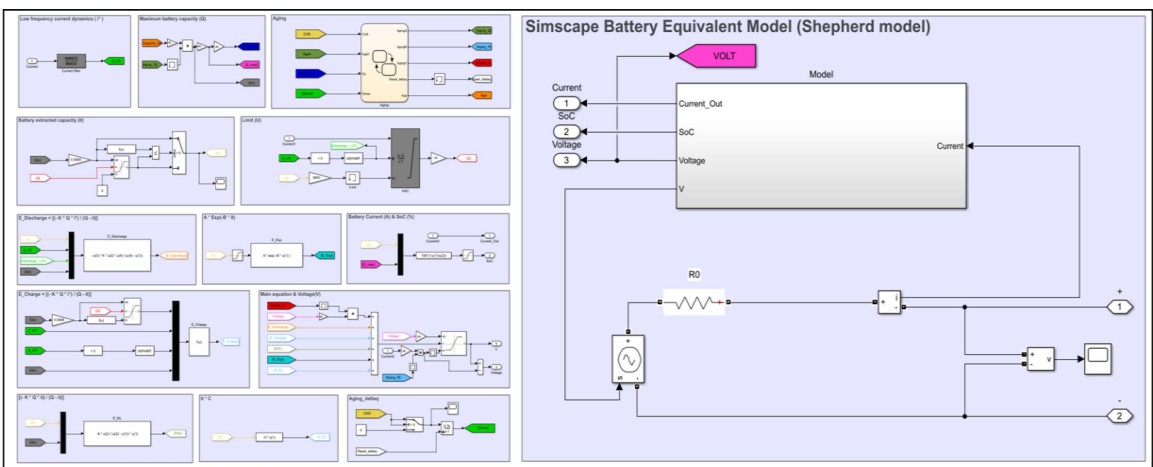

**Figure 5.** Equivalent circuit model based on the Shepherd model.

## 4. Implementation of the Internal Resistance Tracking Algorithm with Simulated Data

As shown in Figure 6, the battery capacity, SOC, and SOH estimation algorithm based on the novel internal resistance tracking method was implemented in the Matlab Simulink environment. For the algorithm's implementation, simulated data from the Shepherd model were used for convenience.

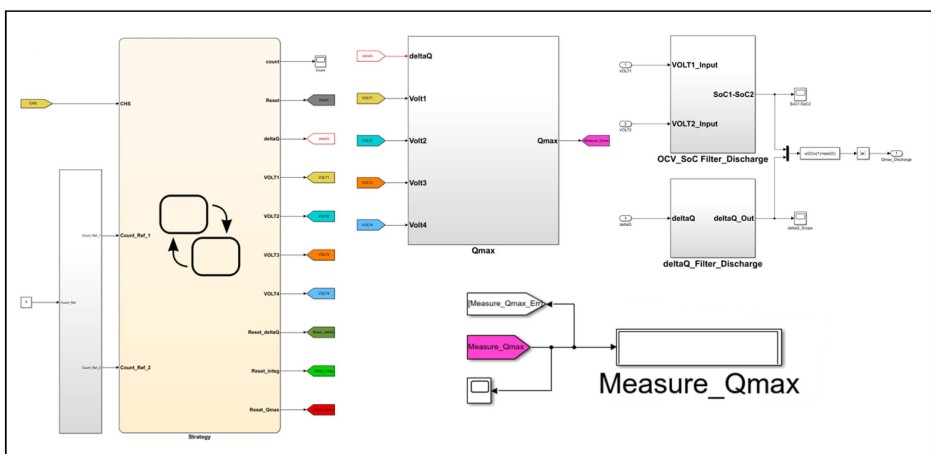

**Figure 6.** Implementation of the new internal resistance tracking algorithm.

The simulations were conducted multiple times based on the battery capacity of 2600 mAh, assuming an environmental temperature of 25 °C. Preliminary analysis based on the simulation data showed that about 500 s are needed to obtain well-relaxed OCV measurement data after the completion of discharging. Therefore, just to be on the safe side, 600 s of relaxation time was assigned to the current profile for cycle tests. As shown in Figure 7a, the current profile consists of a 2 C-rate discharge for 790 s, a relaxation time for 600 s, a 2 C-rate discharge for 650 s, another relaxation time for 600 s, and a final 2 C-rate charge for 1440 s. Cycles were repeated to complete 100 cycles of the simulation.

Various simulation results are plotted in Figure 7a–d, provided above. In the case of the voltage curve, due to the increase in the internal resistance, the voltage drop just before the end of the discharge increased with time. In the SOC curve, as the $Q_{max}$ decreased, the residual SOC decreased, despite the same amount of consumed electricity. The simulation results for the estimation of the internal resistance over cycles are plotted in Figure 8 for Cycles 0, 25, 50, 75, and 100, respectively. During the discharge, the internal resistance changes were compensated accordingly, and the increase in the internal resistance was noticed as it continuously changed.

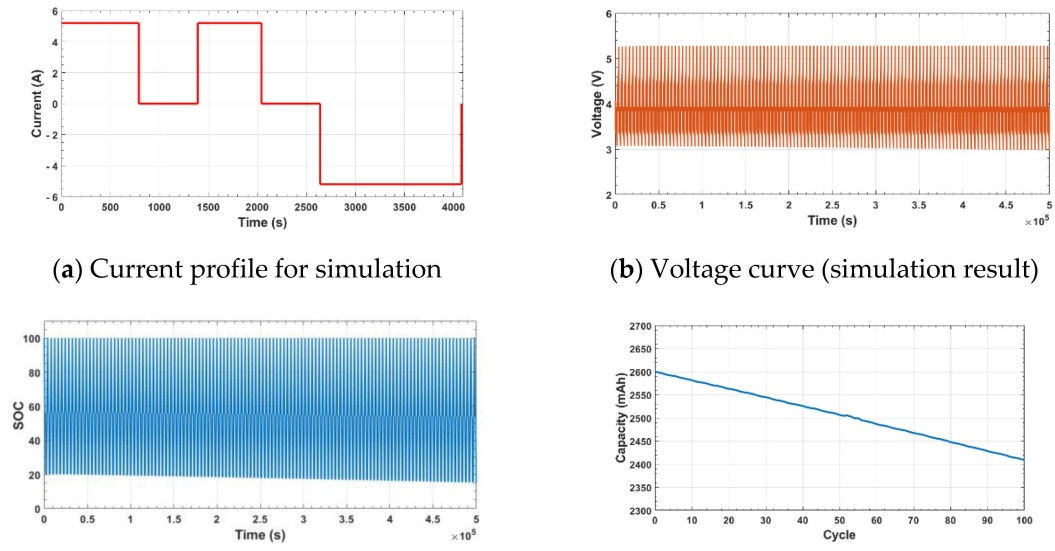

(**a**) Current profile for simulation

(**b**) Voltage curve (simulation result)

(**c**) State of charge (SOC) curve (simulation result)

(**d**) Capacity-cycle curve (simulation result)

**Figure 7.** Current profile and various simulation results.

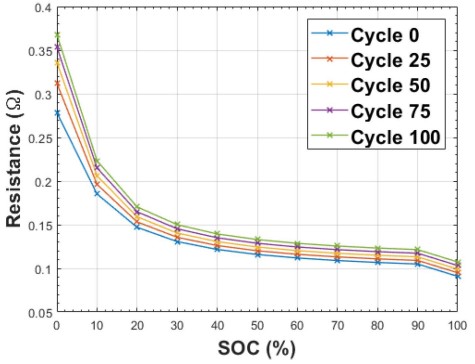

**Figure 8.** Resistance–SOC curve (simulation result).

In addition, the discharge voltage curves are plotted in Figure 9 for Cycles 0, 25, 50, 75, and 100 based on the internal resistance estimation during discharge in real time.

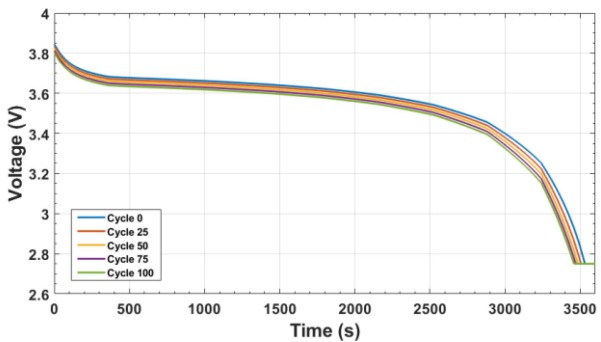

**Figure 9.** Discharge voltage curve for 1 C-rate (2.6 A) (simulation result).

Based on the data shown in Figure 9, the estimated time to reach the compensated end of discharge voltage (CEDV), that is, the end of discharge voltage compensated with the corresponding internal resistance, could be obtained during discharge in real time considering the internal resistance data

respectively for each aging state. As the internal resistance increased, the time to reach the cut-off voltage decreased, and the $Q_{max}$ was also calculated accordingly. The initial 1 C-rate $Q_{max}$ is 9178 Wh, which is smaller than the total available energy (9360 Wh) because Li-ion batteries may discharge to lower voltages than the application-specific cut-off voltage. Most of the time, the cut-off voltage is set to a certain level to enforce the limitation of discharge for battery safety and a stable cycle lifespan. Due to this voltage limitation, the actual usable time of the lithium-ion battery varies depending on application-specific electric current consumption.

As summarized in Table 2, the initial available capacity was 9178 Wh. After 100 cycles, the available energy decreased down to 8988 Wh, resulting in an estimated SOH of 97.9%, which is down by 2.1% from the initial state. With the simulated data on the battery discharging behavior from the Shepherd model, the internal resistance tracking algorithm clearly showed the applicability of the internal resistance estimation method, with the one limitation of a rather long relaxation time, which is about 600 s in the current simulation model but could even be longer in real-life battery applications.

**Table 2.** Calculated time to compensated end of discharge voltage (CEDV), battery capacity ($Q_{max}$), and state of health (SOH).

| Cycles | Cycle 0 | Cycle 25 | Cycle 50 | Cycle 75 | Cycle 100 |
|---|---|---|---|---|---|
| Time to CEDV | 3530 s | 3504 s | 3485 s | 3469 s | 3457 s |
| $Q_{max}$ | 9178 Wh | 9110 Wh | 9061 Wh | 9019 Wh | 8988 Wh |
| SOH | 100% | 99.25% | 98.72% | 98.26% | 97.92% |

## 5. Implementation of the Novel Internal Resistance Tracking Algorithm with Experimental Data

Multiple battery discharge experiments were conducted with an ICR18650-26F Li-ion battery from Samsung SDI to confirm the practical applicability of the newly suggested internal resistance tracking method. In the current investigation, battery charge and discharge experiments were conducted using National Instrument data acquisition devices, an electric load, and a power supply. Current, voltage, and temperature data were collected every second through a PC.

For a better understanding of voltage relaxation, Figures 10 and 11 were prepared. As Figure 10 clearly shows, the relaxation voltage curves behave quite distinctively from each other as the SOC levels—where the discharge stopped at—are rather widely apart (SOC 90%~SOC 0% at every 10%). In contrast to the curves in Figure 10, Figure 11 shows that the relaxation voltage curves exhibit similar patterns to each other as the SOC levels—where the discharge stopped at—are more closely together (SOC 99%~SOC 92% at every 1%).

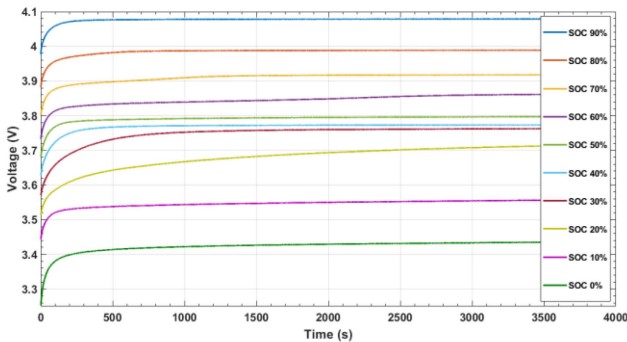

**Figure 10.** Relaxation voltage behavior for SOC 90%~SOC 0% at every SOC 10% step.

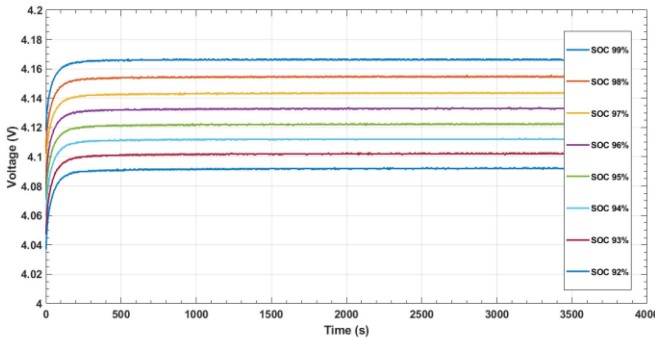

**Figure 11.** Relaxation voltage behavior for SOC 99%~SOC 92% at every SOC 1% step.

With these valuable observations, the novel approach for the fast assessment of $Q_{max}$ was investigated with actual battery data. The main focus of the analysis was to find out how quickly $Q_{max}$ could be evaluated for different situations.

As explained in Section 2.2, the $Q_{max}$ values were estimated with experimentally obtained relaxation voltage curves, where the SOC levels were different from each other by an SOC of 1%. The estimated $Q_{max}$ values were analyzed, and the findings were quite surprising. The estimated $Q_{max}$ values at each SOC level quickly converged and stabilized while the time periods for the converged values were slightly different from each other. Nonetheless, for all cases, the time periods for the convergence were obviously decreased by far compared with those from the conventional internal resistance tracking method. Time periods for the convergence within 1% accuracy of the fully converged $Q_{max}$, which were obtained after 3600 s of relaxation time, are calculated and tabulated in Table 3. As listed in Table 3, asymptotically converged $Q_{max}$ values were obtained in less than 400 s. Considering the fact that the accuracy of SOC measurement at a level of 5% is typically accepted in practical applications, the results of converged $Q_{max}$ values of 1% accuracy within 400 s are excellent.

**Table 3.** Time to measure $Q_{max}$ within 1% accuracy of asymptotically converged $Q_{max}$.

| SOC Levels | 99~98% | 98~97% | 97~96% | 96~95% | 95~94% | 94~93% | 93~92% |
|---|---|---|---|---|---|---|---|
| Time for $Q_{max}$ within 1.00% accuracy of converged $Q_{max}$ | 114 s | 359 s | 247 s | 152 s | 79 s | 139 s | 315 s |
| Time for $Q_{max}$ within 0.50% accuracy of converged $Q_{max}$ | 164 s | 464 s | 394 s | 202 s | 95 s | 196 s | 375 s |
| Time for $Q_{max}$ within 0.25% accuracy of converged $Q_{max}$ | 228 s | 543 s | 549 s | 289 s | 108 s | 268 s | 425 s |
| Time for $Q_{max}$ within 0.10% accuracy of converged $Q_{max}$ | 364 s | 614 s | 691 s | 501 s | 120 s | 606 s | 472 s |

## 6. Conclusions

In this paper, we proposed a novel internal resistance tracking method and demonstrated its successful implementation with experimental data. With this rapid assessment method for battery capacity, resulting in the estimation of SOC and SOH, it is expected that the algorithm can be implemented for real-time measurement in on-board systems. Furthermore, with this newly proposed algorithm, a rapid and viable diagnostic tool for the evaluation of the energy storage capacity of battery packs in retired EVs is fully expected to be developed in the near future. In the meantime, the following remarks are a summarization.

(1) The successful implementation of the newly proposed algorithm was illustrated for efficient use. Based on an investigation with experimental battery data, battery capacity could be measured

within a few minutes instead of 2 h or more, which is typically required by the conventional measurement technique.

(2) While only the simulated data were used to develop the algorithm for the estimation of battery capacity, SOC and SOH based on the newly proposed internal resistance tracking algorithm and experimentally obtained battery data successfully illustrated the applicability of the new algorithm.

(3) In an effort to understand the behavior of voltage curves during the relaxation period, multiple experiments and observations were repeated. Interestingly enough, voltage recovery patterns during the relaxation period showed some distinctive features at different SOC levels. However, a clear similarity was noticed such that the voltage recovery patterns were very close to each other as long as they started at the same SOC, regardless of discharge rate. This observation is quite important to move onto the next level of enhancement of the newly proposed algorithm introduced in this paper.

**Funding:** This research received no external funding.

**Acknowledgments:** The author would like to acknowledge the support from Yonghun Lee with the preparation of figures.

**Conflicts of Interest:** The author declares no conflict of interest.

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
