# Peer review of "A Study on State of Charge and State of Health Estimation in Consideration of Lithium-Ion Battery Aging"

_sustainability, doi:10.3390/su122410451_

Round 1

Reviewer 1 Report

In this paper, conventional methods for the estimation of SOC and SOH was briefly reviewed along with the recently developed internal resistance tracking method. One of the battery models called Shepherd model was constructed based on the experimentally obtained data from a commercially available LIB. After the construction of the battery model, multiple simulations were repeated to produce the discharge curves. With this simulated data from the model, newly suggested SOC and SOH estimation technique was developed and tested. The new method for the SOC and SOH was well established and demonstrated based on the simulated results and the future research plan with real battery data was briefed reasonably.

However, it would like to see a little more detailed description for the next research plan with the real data from the battery at least in the concluding section.

I also found a few mistakes in English, I would like the authors to go through further proof-reading of the manuscript.

Author Response

First of all, I really appreciate careful review from Reviewer 1. Based on the reviewer’s comments, I have prepared a revised manuscript.

Comment 1

In this paper, conventional methods for the estimation of SOC and SOH was briefly reviewed along with the recently developed internal resistance tracking method. One of the battery models called Shepherd model was constructed based on experimentally obtained data with a commercially available LIB. After the construction of the battery model, multiple simulations were repeated to produce the discharge curves. With this simulated data from the model, newly suggested SOC and SOH estimation technique was developed and tested. The new method for the SOC and SOH was well established and demonstrated based on the simulated results and the future research plan with real battery data was briefed reasonably. However, it would like to see a little more detailed description for the next research plan with the real data from the battery at least in the concluding section.

Response 1

Based on the comment from Reviewer 1, I have revised Section 4, Section 5 as follows.

Comment 2

I also found a few mistakes in English, I would like the authors to go through further proof-reading of the manuscript.

Response 2

I have reviewed the manuscript thoroughly and made a few corrections to make it easier to read and to understand.

I really appreciate the thoughtful review from Reviewer 1.

Reviewer 2 Report

56 - Explain in detail why non of the method mentioned cannot be implemented on board measurement device.

79 - The novelty of the paper is just in reducing the time?

Can you explain more and give references why this topic was not focused more by other researchers?

And what's more new in this method.

Need more detailed literature review to make your work more impact.

Do comparison with other method timing and yours and mention a table in the results section or describe in conclusion how your new method improves in terms of timing which is the main focus of the paper.

Author Response

Response to the comments from Reviewer 2

First of all, I really appreciate careful review from Reviewer 2. Based on the reviewer’s comments, I have revised accordingly as follows.

Comment 1

56 - Explain in detail why none of the method mentioned cannot be implemented on board measurement device.

Response 1

I truly appreciated the comment from Reviewer 2. I have added a few paragraphs to accommodate the comment as follows in Section 2.1.

Line 97-101

With the result from Equation (1), Qmax can be compared to the initial quantity at the beginning of life (BOL). As the battery experiences aging, Qmax decreases as a result [20, 21]. Effective metric of the SOH can be establish quickly such as Qmax (now) / Qmax (BOL). The only issue in this concept is a rather long relaxation period which may takes about an hour or more for the stable OCV measurement.

Line 117-123

Regarding conventional SOH estimation techniques, electrochemical impedance spectroscopy (EIS) is the most popular choice among researchers which measures the impedance of the battery by injecting small amount of alternate current (AC) to the battery over wide range of frequencies typically from 0.1 Hz to 1000 Hz to the battery [24, 25]. However, none of these methods can be implemented in the on-board devices for real-time measurements because the EIS requires an off-line setup for the expensive EIS measurement system and the existing internal resistance tracking method requires long relaxation time for satisfactory voltage stabilization.

Comment 2

79 - The novelty of the paper is just in reducing the time? Can you explain more and give references why this topic was not focused more by other researchers?

Response 2

I would like to thank Reviewer 2 for point out the novelty of the current method. In Section 2,.1, I have added a couple of sentences to accommodate the comments from Reviewer 2.

Line 75-90

Typical SOC measurement techniques include a voltage based estimation method, a coulomb counting estimation method, and an internal resistance tracking method [15-23]. Since the first and the second methods are quite commonly used in the field even with the lack of consideration about battery aging effect, they are not detailed in this paper. However, the third technique will be explained in detail as it is the basis for the newly proposed method.

The internal resistance tracking method consists of battery capacity estimation, the voltage-based method and the coulomb counting method together to track internal resistance and eventually to calculate SOC and SOH of the battery considering the aging effect [23]. As shown in Fig. 2, a discharging process is illustrated to explain the basic idea. After the initial discharging process of any shape (from 0 sec to 2000 sec), the battery is going through the first relaxation period. After the reasonable relaxation, OCV at that time is measured and denoted as OCV1. Then, SOC of the battery is obtained using the voltage based method and denoted as SOC1. After that, the second discharging process is carried out. During the second discharge, the amount of discharged energy is measured with the coulomb counting method and denoted as ΔQ. Then, the battery goes through the second relaxation period for the measurement of OCV and denoted as OCV2 resulting in the estimation of SOC using voltage-based method and marked as SOC2, eventually.

Line 127-131

In Equation (1), many researchers paid close attention to obtain both SOC1 and SOC2 separately. For this matter, it usually took long relaxation time to obtain stable SOC1 and SOC2 independently. However, in the current research, a new approach is introduced to overcome this shortcoming. In principle, the difference of SOC1 and SOC2 was directly obtained rather quickly without long relaxation time periods.

Comment 3

And what's more new in this method.

Response 3

Please refer to the Response 2 in the above and the followings.

Line 142-150

In the newly proposed approach, the following concept is introduced. Since the important value for successful estimation of Qmax is the difference between SOC1 and SOC2, namely SOC1-SOC2, not the absolute values of SOC1 and SOC2, a new measurement points are taken. The estimation of SOC1-SOC2 from the respective OCV values are obtained after a reasonably short time instead of waiting for a long relaxation time. In Fig. 3, it is noted in black arrows and more specifically in this case, the SOC1-SOC2 value is estimated after only 120 seconds. As a proof of concept for the proposed approach, simulated battery data from an equivalent circuit model was heavily used during establishment of new methodology. After the satisfactory results from the initial algorithm development, extensive experimental data was collected to further refine the new methodology.

Also, Section 5 is added to provide more details about experimental data analysis.

Comment 4

Need more detailed literature review to make your work more impact.

Response 4

I fully agree with this comment from Reviewer 2. I have added more references for SOC, SOH and EIS measurement techniques as shown in the revised manuscript.

Comment 5

Do comparison with other method timing and yours and mention a table in the results section or describe in conclusion how your new method improves in terms of timing which is the main focus of the paper.

Response 5

I have revised the conclusion section as follows.

  • The successful implementation of the newly proposed algorithm was illustrated for efficient use. Based on the investigation with the experimental battery data, the battery capacity could be measured within a few minutes instead of 2 hours of more which is typically required by the conventional measurement technique.

I also have added a new section to provide the details of experiments that I have done as well. Please, refer to the revised manuscript.

I DO appreciate the comments and suggestions from Reviewer 2 and I would like to show my sincere gratitude.

Reviewer 3 Report

Detailed comments are included in the appendix.

Author Response

Response to the comments from Reviewer 3

First of all, I really appreciate detailed review report from Reviewer 3. I have tried to answer the questions from Reviewer 3 in this response letter as much as possible and also, I have revised the manuscript accordingly in multiple locations wherever they are appropriate.

General

Comment 1

It is difficult to determine what the purpose of the article is and what it actually contains. As noted in Abstract: "In this study, ideas to estimate the SOC of the Li-16 ion battery is discussed" (line 16-17), this is a presentation of research ideas. The material contained in the manuscript is not a scientific article.

Measurement methodologies should be developed, research conducted, and published against the simulations contained in this manuscript.

Response 1

During the early stage of the research, ideas were developed based on the simulated battery model. After the initial write up of the paper, experimental data were analyzed to support the methodology. Following your comment, actual experimental setup and data analysis are added in the manuscript.

Comment 2

Introduction

  1. The authors write that batteries have various uses. This should be expanded and not limited to electric vehicles (line 33). Also, use in vehicles should be supported by a larger body of literature.

Response 2

I have revised the introduction to provide more applications for batteries. Also, I have added more about the importance of EV by pointing out global warming issue as follows.

Line 28~41

The rapid development of technology and industrialization have triggered heavy use of fossil fuels and resulted in to serious environmental problems including pollutant emissions and global warmings [1-3]. Thus, the research and development of eco-friendly energy utilization technology has been rapidly progressing [4, 5]. Among many industrial areas including power generation, manufacturing facility, housing and transportation, the transportation sector has been identified as one of the major responsible party for pollutant emissions and global warmings [6]. Therefore, research and development in electric vehicles (EV) has been accelerating [7]. Among many EV related researches, an energy storage devise, i.e. battery, has attracted the major attention since the battery plays important roles not only in EV but also in mobile devices, personal transportations, energy storage systems (ESS) for renewable energy generation and more. Naturally, research on the battery has been accelerating as well [7-9]. In addition, a battery management system (BMS), which is used for an efficient operation and management of batteries, is also considered as an important system in various applications with high energy storage capacity including electric vehicles and energy storage systems [10-12].

Comment 3

  1. The environmental effects of the use of electric vehicles should be discussed more broadly, as it is not an explicit topic.

Response 3

I have revised the introduction to highlight the effects more broadly as shown in Response 2.

I really appreciate the comment from Reviewer 3

Comment 4

  1. Is the short battery life a major problem? What waste (how much) is generated by the use of electric cars? Does it pose a threat to the environment? Will the authors' research allow extending the life of the battery? How is the authors' research supposed to contribute to the improvement of battery life? Will they reduce the risk to the environment?

Response 4

With the recent development of the Li-ion battery, especially in the case of EV, battery life is actually quite long as far as the battery is used properly. So, the short battery life is not the focus in the current study. Main focus of the current research is to come up with a new methodology to diagnose the battery capacity when the battery is returned for second use after EV retires. For example, conventional diagnostic methods take quite long time up to 10 hours or even more. However, the newly proposed methodology completes the test within 30 minutes or less. With this fast diagnostic capability, battery reuse process will make sense in terms of profitable business activities and as a result, battery resource can be reused with a self-sustaining business model. Without the fast diagnostic techniques, battery for second use can only be discussed but cannot be realized. This is the main contribution of the current research.

So, I have revised the introduction to explain the main focus of the current research as follows.

Line 45~59

With this increase in EV sales, it is natural to expect that the number of EV retired from service will increase in the similar matter with a time delay of 10 or more years. With the recent development of the Li-ion battery, especially for the case of EV, battery life is actually quite long as far as the battery is used properly. So, the short battery life is the main issue. Main focus of the current research is to come up with a new methodology to diagnose the battery capacity, typically denoted as Qmax, when the battery is returned for second use after EV retires. For example, conventional diagnostic methods take quite long time up to 10 hours or even more. However, the newly proposed methodology would complete the test within 30 minutes or less. With this fast diagnostic capability, battery reuse process will make sense in terms of profitable business activities and as a result, battery resource can be reused with a self-sustaining business model. Without the fast diagnostic techniques, battery for second use can only be discussed but cannot be realized. Furthermore, only with this updated battery capacity, SOC and SOH can be estimated accurately as the battery ages more and more. Therefore, newly proposed method for the fast measurement of battery capacity is the main contribution of the current research.

Comment 5

  1. The second paragraph of the Introduction (lines 37-45) is methodological. Important information from this part should be transferred to a clearly separated chapter containing the research objective, scope, and own research methods.

Response 5

I have revised the manuscript to accommodate the comment from Reviewer 3 as follows.

  1. Methodology Development

Newly proposed idea for the fast estimation of battery capacity was developed by adopting the internal resistance tracking method. After the establishment of the initial idea, an equivalent circuit model of a specific Li-ion battery was established based on a Shepherd model that would mimic open circuit voltage (OCV) characteristic behaviors very well [14, 15]. Main reason for constructing the battery model was to produce numerous simulation data for the successful implementation of the new methodology. With the established model, multiple simulations were carried out to produce quantitative data which clearly showed the increase of internal resistance and the decrease of battery capacity as the battery aged. With the help of simulation data, the newly proposed approach was refined and after the successful implementation of the new methodology, real battery experiments were conducted to prove the concept of fast measurement of the battery capacity [16]. In the following sections, conventional methods and the newly proposed method are explained.

Comment 6

Chapters 2-4

Chapters 2-4 describe the different methods of assessing SOC and SOH, but without even providing the basis of the literature. Therefore, it is neither a scientific nor a review article.

Response 6

Following the guide from Reviewer 3, number of reference materials were added accordingly in Chapter 2 as shown below.

  1. Methodology Development

Newly proposed idea for the fast estimation of battery capacity was developed by adopting the internal resistance tracking method. After the establishment of the initial idea, an equivalent circuit model of a specific Li-ion battery was established based on a Shepherd model that would mimic open circuit voltage (OCV) characteristic behaviors very well [14]. Main reason for constructing the battery model was to produce numerous simulation data for the successful implementation of the new methodology. With the established model, multiple simulations were carried out to produce quantitative data which clearly showed the increase of internal resistance and the decrease of battery capacity as the battery aged. With the help of simulation data, the newly proposed approach was refined and after the successful implementation of the new methodology, real battery experiments were conducted to prove the concept of fast measurement of the battery capacity. In the following sections, conventional methods and the newly proposed method are explained.

2.1 Conventional Methods for SOC Measurement

Typical SOC measurement techniques include a voltage based estimation method, a coulomb counting estimation method, and an internal resistance tracking method [15-21]. Since the first and the second methods are quite commonly used in the field even with the lack of consideration about battery aging effect, they are not detailed in this paper. However, the third technique will be explained in detail as it is the basis for the newly proposed method.

Comment 7

  1. 103-105

Why was the experimental data not used? On line 171 there is information about many of the authors' experiments.

Response 7

Please refer to the Response 1

Comment 8

  1. 116-117

In the previous paragraph (line 104) it was mentioned that the data from the model were used instead of the experimental data. Here, in turn, there is information that experimental data were used to verify the model. Where is this data in the manuscript?

Response 8

In order to accommodate the comment from Reviewer 3, Section 5 has been added. Please refer to the manuscript.

Thank you very much.

Comment 9

Discussion

Has no one done research on this? There is no discussion in the article with the results of other studies.

Response 9

I have added a few more reference to highlight the issue.

Comment 10

Conclusions

The conclusions do not relate to the content of the manuscript.

  1. 173

Who and in what package will the data from the research results be made available?

  1. 173

What conference are the authors writing about? Do these conclusions really apply to the submitted manuscript?

  1. 179, 181, 185

In their conclusions, the authors repeatedly refer to the numerous studies carried out, which they did not present in the manuscript.

Response 10

A few typos were corrected and the details of the experimental data and related analysis were added in Section 5. Please, refer to the revised manuscript.

I DO appreciate the comments from Reviewer 3 for many suggestions and guides.

Comment 11

References

The authors used only 10 items of literature, two of which are product characteristics. The topic covered in the manuscript is actual and you can find a lot of research on different aspects of the topic (technical, environmental). A full analysis of the current situation in this field should be made, based on the scientific literature. A bibliography should also be prepared in accordance with the requirements of the journal.

Response 11

I have added a few more references in the paper to accommodate the request from Reviewer 3.

Technical Notes

Comment 12

Give explanation of abbreviations where they are used for the first time.

Response 12

Explanations were added wherever possible. Thank you for the clarification.

Comment 13

The formulas, symbols and abbreviations used in them should be explained, as well as units.

Response 13

Based on the comment, I have revised the manuscript accordingly.

Comment 14

The source of figures and data in the tables should be provided. Currently, it is

possible to guess, based on the analysis of the text, that some of the figures come from the given literature sources, but this information should be clearly stated under each graph and table.

Response 14

I have added the source of the data where it is needed accordingly. Again, Thanks for the guide to improve the manuscript.

Comment 15

Their captions should not be placed on the graphs, but in the description below. In its current form, each drawing has two captions that are not always even consistent.

Response 15

I have revised the figures to reflect the correction pointed out by Reviewer 3.

I DO APPRECIATE YOUR KIND AND CAREFUL REVIEW FOR MY MANUSCRIPT.

I THANK YOU VERY MUCH.

Round 2

Reviewer 3 Report

The manuscript has been revised according to my suggestions. Thanks to adding research results, supplementing references and organizing the content, its quality and importance has definitely increased.

The article may be published in present form.